## SCIENCE FORUM

# The single-cell eQTLGen consortium

**Abstract** In recent years, functional genomics approaches combining genetic information with bulk RNA-sequencing data have identified the downstream expression effects of disease-associated genetic risk factors through so-called expression quantitative trait locus (eQTL) analysis. Single-cell RNA-sequencing creates enormous opportunities for mapping eQTLs across different cell types and in dynamic processes, many of which are obscured when using bulk methods. Rapid increase in throughput and reduction in cost per cell now allow this technology to be applied to large-scale population genetics studies. To fully leverage these emerging data resources, we have founded the single-cell eQTLGen consortium (sc-eQTLGen), aimed at pinpointing the cellular contexts in which disease-causing genetic variants affect gene expression. Here, we outline the goals, approach and potential utility of the sc-eQTLGen consortium. We also provide a set of study design considerations for future single-cell eQTL studies.

**MGP VAN DER WIJST[†]\*, DH DE VRIES[†], HE GROOT, G TRYNKA, CC HON, MJ BONDER, O STEGLE, MC NAWIJN, Y IDAGHDOUR, P VAN DER HARST, CJ YE, J POWELL, FJ THEIS, A MAHFOUZ, M HEINIG AND L FRANKE**

## Interindividual variation needs to be studied at the single-cell level

Genetic variants, most commonly single nucleotide polymorphisms (SNPs), can contribute to disease in a plethora of ways. In monogenic diseases, one single variant is sufficient to result in a disease phenotype. In complex diseases, tens to hundreds of variants each independently contribute to disease risk and an accumulation of risk alleles – often in combination with specific environmental exposures – is required to develop the disease phenotype. The overwhelming evidence showing enrichment of disease-associated variants in regulatory regions suggests that regulation of gene expression is likely a dominant mediator for disease risk. Expression quantitative trait loci (eQTL) analysis is commonly used for linking disease risk-SNPs to downstream expression effects on local (*cis*) or distal (*trans*) genes. Large-scale eQTL efforts such as GTEx (*GTEx Consortium, 2017*), PsychENCODE (*Wang et al., 2018*), ImmVar (*Ye et al., 2014*), BLUEPRINT (*Chen et al., 2016*), CAGE (*Lloyd-Jones et al., 2017*), and eQTLGen (*Võsa, 2018*) have proven highly valuable to identify downstream transcriptional consequences. All these efforts together lead to ever growing sample sizes that now allow us to start identifying both *cis*- and *trans*-eQTLs.

An important next step is to precisely define the cellular contexts in which disease risk-SNPs affect gene expression levels. This will help to better understand the molecular and cellular mechanisms by which disease risk is conferred and to inform therapeutic strategies. This needs to be done as recent analyses have shown that many eQTL effects are tissue (*GTEx Consortium, 2017*; *Fu et al., 2012*) and cell type-specific (*Brown et al., 2013*; *Fairfax et al., 2012*). Additionally, many eQTLs are conditional, and only revealed after specific stimuli that, for example, change the activation or differentiation of specific cell types (*Ye et al., 2014*; *Cuomo et al., 2020*). Beyond the ability to annotate individual disease associations, cell type-specific eQTLs have been shown to be strongly enriched for heritability across complex traits (*Hormozdiari et al., 2018*). Sorting (*Fairfax et al., 2012*; *Ishigaki et al., 2017*) and computational deconvolution (*Westra et al.,*

**\*For correspondence:** m.g.p.van.der.wijst@umcg.nl

[†]These authors contributed equally to this work

**Competing interests:** The authors declare that no competing interests exist.

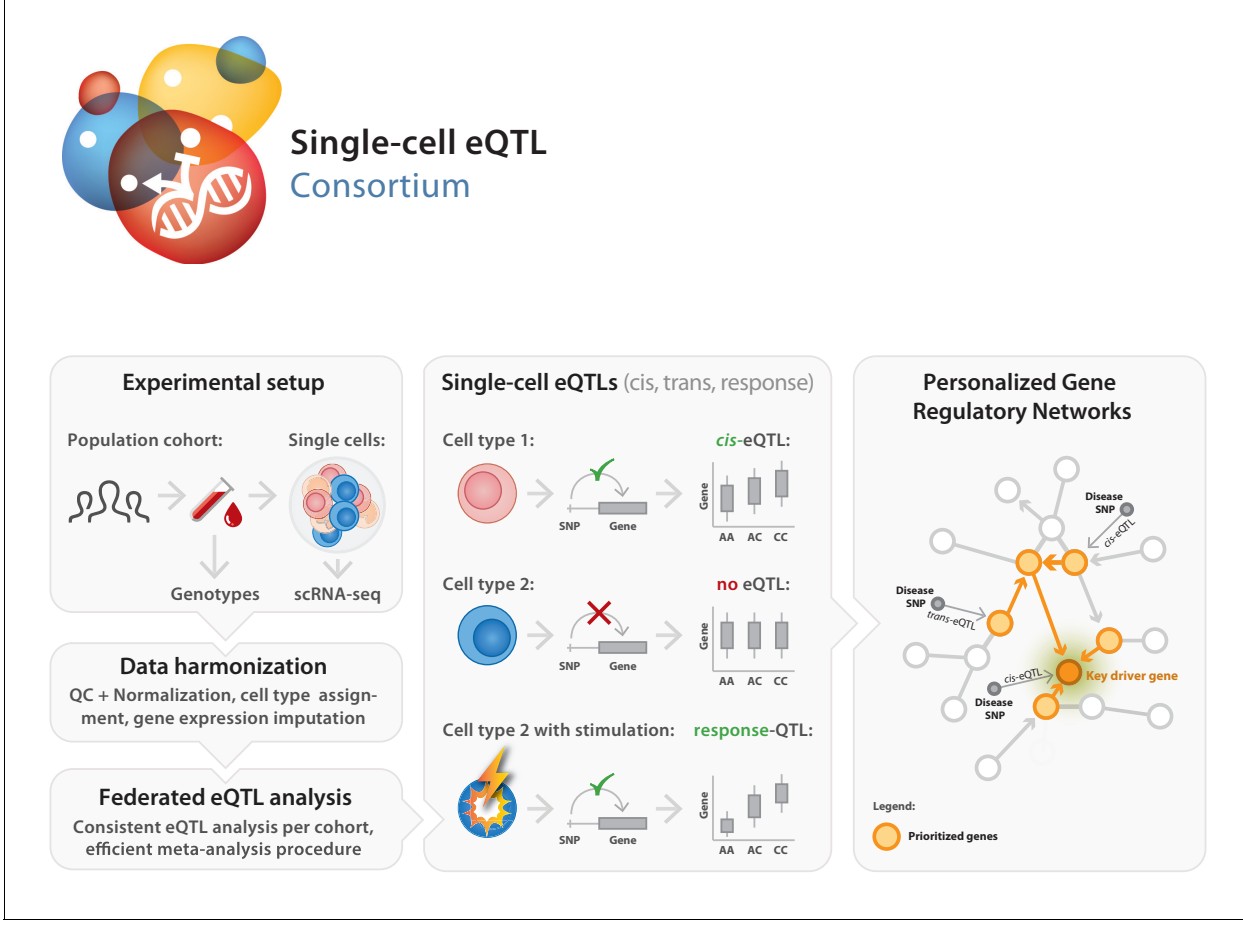

**Figure 1.** Set-up of the single-cell eQTLGen (sc-eQTLGen) consortium. The sc-eQTLGen consortium combines an individual's genetic information with single-cell RNA expression (scRNA-seq) data of peripheral blood mononuclear cells (PBMCs) in order to identify effects of genetic variation on downstream gene expression levels (eQTLs) and to enable reconstruction of personalized gene regulatory networks. Right panel is adapted from *van der Wijst et al. (2018b)*.

*2015*; *Venet et al., 2001*) of cell types from bulk samples have been used to uncover context-specificity of eQTLs. However, these methods are biased towards known cell types defined by a limited set of marker genes (*Zhernakova et al., 2017*), are of limited use for less abundant cell types, and do not capture any heterogeneity within a sorted population. In contrast, single-cell RNA-sequencing (scRNA-seq) enables the simultaneous and unbiased estimation of cellular composition and cell type-specific gene expression (*van der Wijst et al., 2018a*), and is particularly well positioned to investigate rare cell types (*Villani et al., 2017*). As opposed to using bulk data, single-cell data allows us to also link genetics to phenomena such as cell-to-cell expression variability (*Cuomo et al., 2020*), cell type heterogeneity (*Donovan et al., 2019*), and gene regulatory network differences (*van der Wijst et al.,*

*2018a*). As such, single-cell analyses in a population-based setting will likely become mainstream in the next few years. However, we envision that most scientific value will be obtained by unifying these efforts. Additionally, to utilize the aforementioned developments in the single-cell field most efficiently and effectively, a coordinated effort from multiple research groups is urgently needed.

Here we introduce the single-cell eQTLGen consortium (sc-eQTLGen), a large-scale, international collaborative effort that has been set up to identify the upstream interactors and downstream consequences of disease-related genetic variants in individual immune cell types (https://eqtlgen.org/single-cell.html, *Figure 1*). In this consortium we will attain a sufficiently large sample size to have the statistical power to unbiasedly identify cell type-specific effects on both *cis* and *trans* genes. Moreover, we aim to

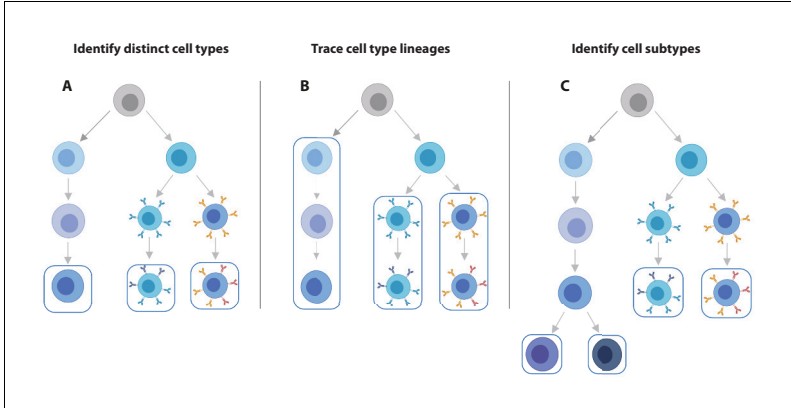

**Figure 2.** scRNA-seq data offers increased flexibility in the eQTL analysis strategy over bulk RNA-seq data. Using scRNA-seq data for eQTL mapping offers a number of advantages over bulk RNA-seq based approaches, of which the flexibility in analysis strategy is a major one. (A) From single cell data, individual cell types can be identified and we can map eQTLs for each of these. (B) Alternatively, lineages based on either knowledge of cell developmental lineages or through pseudo-time based approaches can be constructed. By positioning cells across a trajectory dynamic changes in the allelic effects on gene expression levels as a function of trajectory position can be integrated. (C) Finally, as the discoveries of new cell subtypes are made or cell type definitions are being refined, the analysis can be revisiting by re-classifying cells and determining how the genetic effects on gene expression vary on these new annotations.

reconstruct context-specific gene regulatory networks (GRNs) by combining single-cell and bulk RNA-seq datasets for increased resolution. We expect the results of sc-eQTLGen to have an impact in a number of areas including the prioritization of disease-risk genes, the prediction of drug efficacy and the reconstruction of personalized GRNs.

## Integration of sc-eQTLGen within the scientific landscape

Large numbers of single cell expression profiles from many individuals are required to reach our goals. The accessibility and clinical relevance of peripheral blood mononuclear cells (PBMCs) have made them the most studied cell types in current population-based scRNA-seq datasets. Therefore, to have such datasets from the same tissue type readily available, we have chosen to focus on PBMCs. It also allows for continuation of the knowledge acquired through the eQTLGen consortium, which performed the largest eQTL meta-analysis to date using whole blood bulk gene expression data of over 30,000 individuals to reveal the influence of genetics on gene expression (*Võsa, 2018*). The sc-eQTLGen consortium now allows us to take the next step by systematically assessing the cell types and contexts in which the eQTL effects manifest.

Beyond resolving the influence of genetics on individual genes, the consortium will also take advantage of the unique features of scRNA-seq data to learn the directionality of GRNs and uncover how genetics is affecting co-expression relationships (*van der Wijst et al., 2018a*). We expect that the infrastructure and best practices developed within sc-eQTLGen can serve as a basis for studying population genetics at the single-cell level in solid tissues in the future.

Other large-scale efforts such as the Human Cell Atlas (HCA) (*Regev et al., 2017*) or Lifetime FET flagship consortium (https://lifetime-fetflagship.eu) mainly focus on mapping all cells of the human body or a disease context in a limited number of individuals. The sc-eQTLGen consortium will complement those efforts by putting a unique focus on deciphering the impact of genetic variation on gene expression and its regulation. Different to experimental designs that aim to generate an extensive map on a low number of individuals, we require larger numbers of individuals, whereas the number of cells per individual can be lower. This will enable the accurate capture of both the genetic variation and the cell type heterogeneity. By building on the data and harmonized cell type annotations generated within the HCA, our results will be easily transferable to other datasets as well. We will share best practices of the HCA consortium with regard to data acquisition, analysis and reporting. We also share standards for open science and the infrastructure and legal frameworks for data sharing while accounting for the privacy issues specific to genetic, health record and demographic information.

## Single cell eQTL analysis: the new era of population genetics

The practice of identifying eQTLs is shifting from bulk to single-cell analyses. Considering only its ability to identify eQTLs, scRNA-seq data has a lower statistical power compared to bulk RNA-seq data on the same number of donors (*Cuomo et al., 2020*; *Sarkar et al., 2019*), likely due to increased sparsity of the single-cell data. Nevertheless, there are several clear benefits of single-cell over bulk expression data for QTL analysis. First, scRNA-seq data enables the simultaneous estimation of the composition and expression profiles of discrete cell populations including cell types and their activation states (*van der Wijst et al., 2018a*; *Figure 2*). Second, scRNA-seq data provides a flexible, unbiased approach that has increased resolution to define

## Box 1. Guidelines for creating a population-based single-cell cohort.

Even though a single-cell eQTL dataset has less discovery power than an equal-sized bulk RNA-seq eQTL dataset (6.9 fold difference based on the lowest correlation that led to the identification of a significant eQTL from single-cell [*van der Wijst et al., 2018a*] vs bulk RNA-seq data [*Zhernakova et al., 2017*]), it does provide insights that cannot easily be extracted from bulk data. For example, single-cell data allows for the unbiased detection of cell type- and context-dependent eQTLs and has more power to detect co-expression QTLs (*van der Wijst et al., 2018a*). This makes population-based single-cell datasets a valuable addition to bulk-based datasets for studying the effects of genetic variation on gene expression and its regulation (*van der Wijst et al., 2018a*; *Kang et al., 2018*). In comparison to 'standard' single-cell datasets, generating such population-based single-cell datasets require some additional aspects to be taken into account.

First of all, the genetic information that is available for each of the individuals in such cohorts can be used to demultiplex pools of multiple individuals within the same sample. This approach allows to properly randomize experiments, while also significantly reducing cost and confounding effects (*Kang et al., 2018*; *Bycroft et al., 2018*). This genetic information can either be efficiently generated using genotype arrays (*Marchini and Howie, 2010*) in combination with imputation-based approaches (*Xu, 2019*), or extracted from the scRNA-seq data itself (*Bycroft et al., 2018*; *McCarthy et al., 2016*). Within the consortium all reads will be aligned to the GRCh38/hg38 reference genome and genotypes will be imputed using the Haplotype Reference Consortium reference panel (*Gravel et al., 2011*). The basic principle behind genetic multiplexing is that enough transcripts harboring SNPs are expressed and detected in each single cell such that cells can be accurately assigned to the donor of origin. Furthermore, as the number of multiplexed individuals increases, the probability that a droplet harbors multiple cells from different individuals increases, thus allowing the detection of multiplets using genetic information. This enables the overloading of cells into standard droplet-based workflows and overall reduction of cost per cell up to about 10-fold (https://satijalab.org/costpercell). As the cost of sequencing and the background multiplet rate reduce, the benefits of multiplexing increase. We anticipate that future workflows will allow for even higher throughput.

Secondly, accounting for ethnicity variation and population stratification will be required when single-cell data of diverse populations are being analyzed. It is known that a different genetic architecture exists between different populations. Nevertheless, practical considerations have limited the majority of eQTL studies to cohorts of European origin. As an undesirable consequence of this bias in population representation, certain variants may not have been detected before (*Carlson et al., 2013*) or the effect sizes and associated polygenic risk scores based on the European population may not be translatable to other populations (*Martin et al., 2017*; *Sirugo et al., 2019*). Therefore, inclusion of datasets from different ethnic populations will help reduce long-standing disparities in genetic studies and has many analytical advantages (*Wojcik et al., 2019*; *Hsiao et al., 2010*). For example, the increased genotype frequency diversity will enhance the range over which gene expression varies, and thereby, will further increase statistical power. To implement multi-population sc-eQTL analysis, several challenges have to be addressed. Handling data from populations with different levels of population genetic properties such as LD structure, relatedness and multiple genetic origins that result in the presence of genetic covariance remains important and requires appropriate adjustments to avoid spurious signal and to manage the bias in estimating genetic *cis*- and *trans*-effects (*Zhou and Stephens, 2012*; *Mandric, 2019*). This is particularly important when differences in cohort-specific genetic characteristics are enhanced such as when family-based and unrelated cohorts or cohorts of different ancestries are analyzed. Failing to account for these effects affects the accuracy of mapping and results in false positives.

Finally, studying genetic variation at the single-cell level adds some extra requirements for the number of cells per individual and the number of individuals to be included in the study. The number of cells per individual will mainly define for which cell types in a heterogeneous sample such as PBMCs eQTL and co-eQTL analyses can be performed. In contrast, the number of individuals will mainly define the number of genetic variants for which effects on gene expression can be confidently assessed. A recent analysis showed that, with a fixed budget, the optimal power for detecting cell type-specific eQTLs is obtained when the number of reads is spread across many individuals[119]. Even though a lower sequencing depth per cell results in a lower accuracy of estimating cell type-specific gene expression levels, many more individuals and cells per individual can be included for the same budget. As a result, the optimal experimental design with a fixed budget provides up to three times more power than a design based on the recommended sequencing depth of 50,000 reads per cell (for 10X Genomics scRNA-seq). In contrast, for co-eQTL analysis there is a different trade-off between sequencing depth, number of individuals and number of reads per cell; while for eQTL analysis gene expression levels among cells of the same cell type can be averaged, for co-eQTL analysis you cannot as this would prohibit you from calculating a gene-gene correlation per individual. Therefore, for co-eQTLs the sequencing depth will be a major limiting factor that determines the number of genes for which you can confidently calculate gene-gene correlations. Altogether, depending on the goal of your study, the optimal balance between sequencing depth and

number of individuals and cells per individual will be different. By the end of 2020, the sc-eQTLGen consortium will provide standardized pipelines and guidelines for single-cell population genetics studies.

cell states along continuous dynamic processes in which the eQTL effects manifest themselves (*Cuomo et al., 2020*). Third and fourth, single-cell data allows estimating the variability in gene expression across individual cells (*Brennecke et al., 2013*; *Eling et al., 2018*), which could be used to improve mean estimations for eQTL analysis. At the same time, the single-cell nature now also enables us to look at the effect of genetic variation on transcriptomic traits other than average gene expression level, such as dispersion QTLs that alter the variance independently of the mean expression (*Sarkar et al., 2019*) or cell type proportion QTLs (*Kang et al., 2018*), providing a new angle on how genetic variation may impact disease pathogenesis. Fifth, the large number of observations per individual (i.e. cells) enable the generation of personalized co-expression networks, which vastly reduces the number of individuals required to identify SNPs altering co-expression relationships (i.e. co-expression QTLs [*van der Wijst et al., 2018a*]). Finally, and paradoxically, is the potential benefit of lower experimental costs compared to bulk experiments on sorted cells: such experiments require a library to be generated for each sorted population, whereas a single scRNA-seq library of just one sample contains all this information and can easily be multiplexed across multiple individuals (*Kang et al., 2018*).

So far, only a limited number of papers have performed eQTL analysis using scRNA-seq data (*Cuomo et al., 2020*; *van der Wijst et al., 2018a*; *Sarkar et al., 2019*; *Kang et al., 2018*). In the earliest single-cell eQTL studies, bulk-based eQTL analysis approaches, such as Spearman rank correlation (*Heap et al., 2009*; *Stranger et al., 2007*) and linear regression (*Michaelson et al., 2009*; *Stranger et al., 2005*), were applied to the average expression level of all cells from a particular cell type per individual. However, the underlying assumptions of these bulk-based approaches may not be applicable to scRNA-seq data. Therefore, these bulk-based methods will lose statistical power when applied to scRNA-seq data, because of the inflation of zero values (i.e. sparsity). More recently, single-cell-specific eQTL methods have

been developed that, for example, take into account zero-inflated gene expression (*Sarkar et al., 2019*; *Hu and Zhang, 2018*) or take advantage of pseudotime (i.e. statistically inferred time from snapshot data) to increase the resolution by which response-/differentiation-associated eQTLs (dynamic eQTLs, i.e. eQTLs that dynamically change along pseudotime) can be identified (*Cuomo et al., 2020*). Instead of averaging gene expression levels across all cells from a particular cell type, some of these approaches look at the fraction of zero expression and the non-zero expression separately for each gene (*Hu and Zhang, 2018*). Other approaches take dynamic pseudotime-defined instead of statically-defined cell types into consideration for the eQTL analysis (*Cuomo et al., 2020*). This latter approach was shown to uncover hundreds of new eQTL variants during iPSC differentiation that had not been detected when static differentiation time points would have been used (*Cuomo et al., 2020*). In line with this, we expect that some of these methodological advances, as opposed to bulk-based approaches, will further improve the power and resolution of single-cell eQTL analysis. However, there are two initial challenges that need to be carefully addressed for single-cell eQTL mapping: firstly, the normalization of data to remove technical variation in sequencing depth per cell, while avoiding the removal of biological variation; and secondly, the identification or classification of a cell into a cell type or state.

During library preparation and sequencing, technical and stochastic factors will lead to variation in cell-to-cell sequencing depth. However, simply normalizing to equal sequencing depth per cell will remove important biological variation – for example a $CD4^+$ T cell is expected to have lower RNA contents than a plasma B cell. Therefore, we need to employ normalization strategies that can account for traditional batch effects, such as sample run or sequencing lane, while retaining biological differences (*Bacher et al., 2017*; *Hafemeister and Satija, 2019*).

Once normalized, each cell needs to be accurately annotated into a cell type and/or cell state

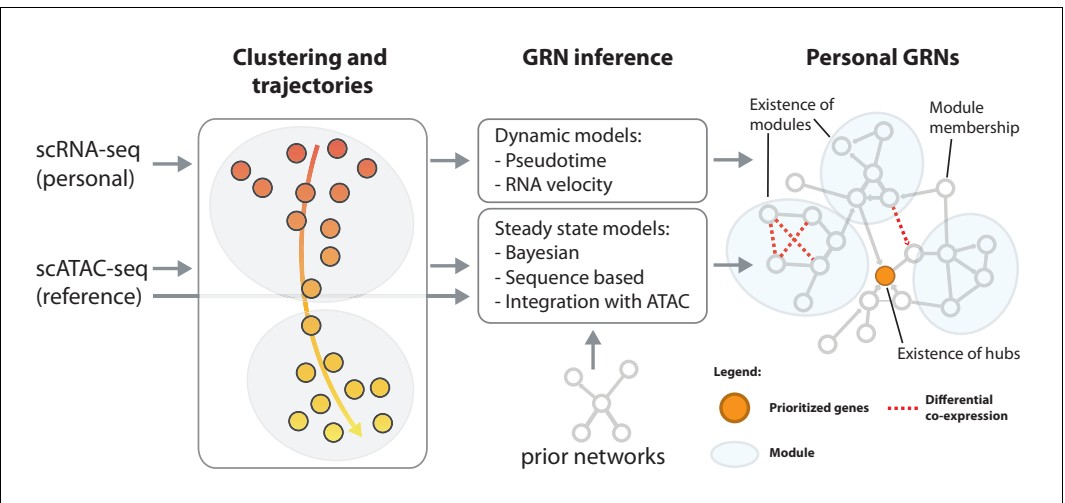

**Figure 3.** Reconstruction of personalized gene regulatory networks. Individual and cell-type specific scRNA-seq data will be used to construct personalized gene regulatory networks. Some single cell datasets allow for the inference of trajectories, for instance in response to a stimulus. These can be used as input to dynamic models to infer causal (directed) interactions. Steady state datasets, characterized by cell type clusters can be analyzed with models that exploit co-expression, prior networks or cell type-specific reference scATAC-seq datasets in combination with sequence motifs to infer directed transcription factor-target relations. Topological comparison between personalized networks of groups of individuals can reveal coordinated differences, for instance the change of connectivity in densely connected modules, change of connectivity of hub genes or changes of module membership of individual genes.

to maximize the statistical power to detect cell type-specific eQTLs. We encourage the use of individual cell classification approaches, rather than cluster-labeling methods. Clustering approaches are powerful ways of identifying a subpopulation of cells that share similar expression levels. However, while most cells placed in a specific cluster will likely be the same cell type, clusters can also contain alternative cell types. Labeling all cells in a cluster based on a high percentage of the expression of a canonical marker(s) will therefore lead to the incorrect classification of some cells (*Alquicira-Hernández et al., 2018*). To acquire a reliable classification model, large scRNA-seq datasets from various contexts are required. Such datasets have been collected within large-scale efforts such as our consortium and the HCA. We expect these will help to develop a gold standard classification model that can classify each cell independently. This will ensure a higher accuracy in cell labeling and thus will maximize power to detect cell type-specific effects.

After solving these challenges, eQTLs can be mapped by either averaging the normalized expression levels on a per gene, per cell type, per individual basis. Alternatively, each cell from an individual can be taken as a repeated measure which can then be used to fit a statistical model to all cells, while including a random effect of the individual.

Instead of using observational studies, eQTLs could also be identified through experimental approaches that use single cells as individual units of experimentation (*Gasperini et al., 2019*). Sample multiplexing (see *Box 1*) can be combined with experimental perturbation to more efficiently characterize the genetic architecture of gene expression. For example, synthetic genetic perturbations with CRISPR/Cas9 may allow precise control of the expression levels of target gene regulators enabling the validation of detected *trans*-eQTLs and the establishment of upper and lower bounds of *trans* effects. Encoding environmental and genetic perturbations across large population cohorts also enables new designs for studying genetic interactions, both gene-by-environment and gene-by-gene (epistasis). Historically, characterizing these effects in human cells has been plagued by the lack of power and the susceptibility to technical confounding of bulk experiments. Recent work that knocked out ~150 regulators in primary human T cells of nine donors illustrates a proof of concept of how single-cell sequencing across individuals can be combined with experimental perturbations to detect these genetic interactions (*Gate, 2019*).

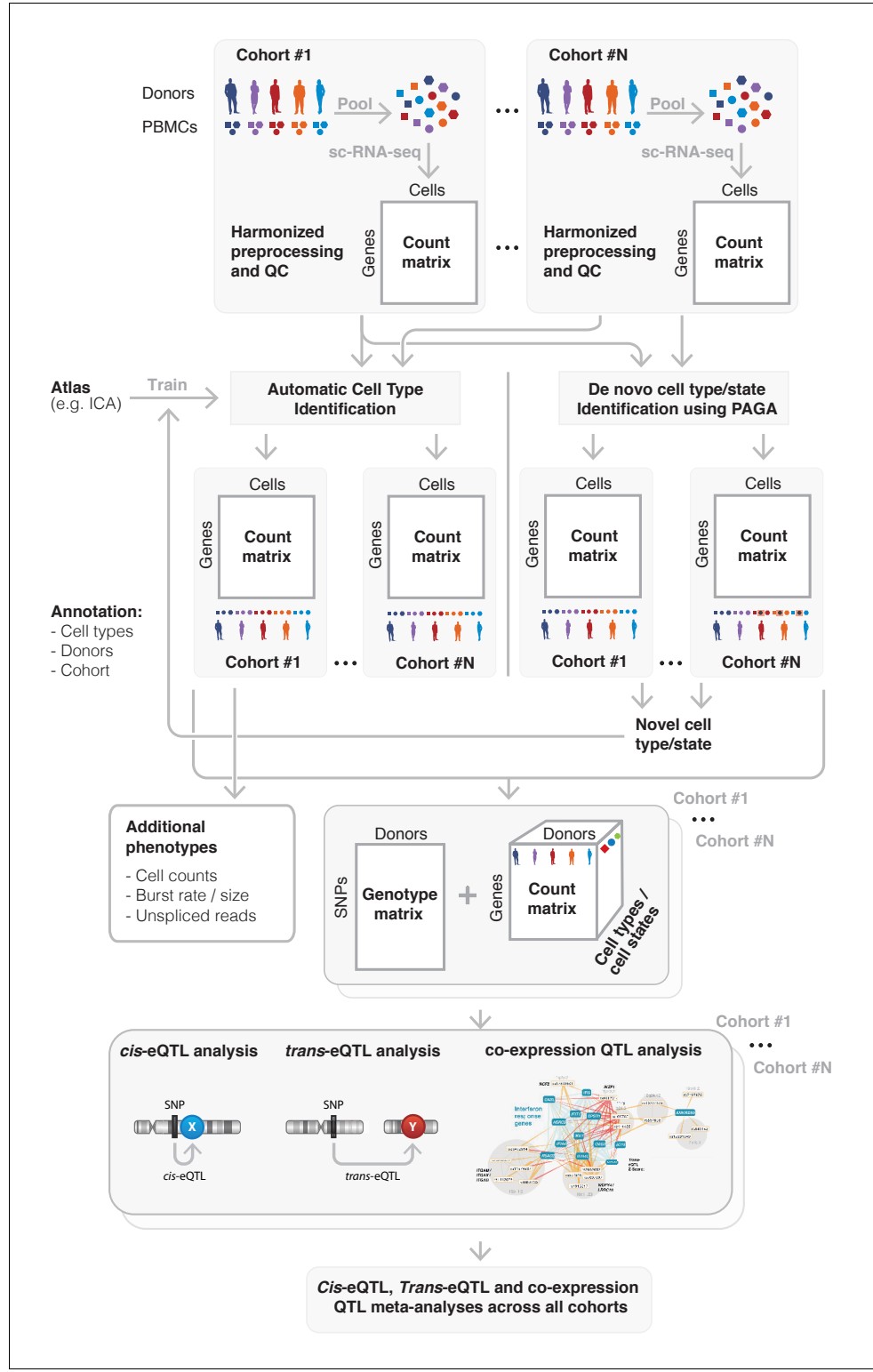

**Figure 4.** Overview of the sc-eQTLGen proposed federated approach. sc-eQTLGen aims to identify the downstream consequences and upstream interactors of gene expression regulation. To increase the resolution and power of this analysis, datasets of multiple cohorts need to be combined while taking privacy issues into account. This will be done using a federated approach in which we will first harmonize all preprocessing and quality control (QC) steps across cohorts. Subsequently, shared gene expression matrices will be normalized and cell types will be classified based on a trained reference dataset (e.g. Immune Cell Atlas (ICA)). Any cells that

*Figure 4 continued on next page*

*Figure 4 continued*
cannot be classified using this trained classifier, representing new cell types or previously unknown cell states, can then be manually annotated based on marker genes, and then be used to further train the classifier. Each cohort will then separately perform a *cis-* and *trans-*eQTL and co-expression QTL analysis using their genotype and expression matrix, while using appropriate statistical models to account for effects such as gender, population structure and family-relatedness that can alter the genotype-expression relationship in a cohort-specific manner. The summary statistics will be shared and analyzed in one centralized place. Finally, these results will be used for reconstruction of personalized and context-specific gene regulatory networks. Bottom panel is reproduced from *Võsa (2018)*.

Another promising avenue that has become available in recent years to gain increased insights in the link between genetics and disease, is through the use of spatial transcriptomics technologies, including MERFISH, seqFISH+, Slide-seq and 10x Visium (*Burgess, 2019*; *Maynard et al., 2020*). While for PBMCs this approach may not be applicable, in solid tissues and organs this extra layer of spatial information is extremely valuable. For example, it can help to disentangle *trans-*eQTL interactions that are modulated through cell-cell communication (e.g. a SNP affects ligand expression in one cell type, and thereby affects downstream receptor signaling in a second cell type). Despite not having this spatial information available in PBMCs, other approaches that consider receptor-ligand expression pairs do provide insights in potential cell-cell interactions. These approaches have been successfully applied before to uncover how the ligand expression in one cell type can affect the frequency (*Smillie et al., 2019*) or the downstream signaling (*Arneson et al., 2018*) of another cell type expressing the corresponding receptor.

## Single cell GRN reconstruction: taking eQTLs one step further

In the case of complex diseases, it is not the disruption of a single gene that causes the disease phenotype. In fact, hundreds of variants can contribute to the disease and converge into just a few key disrupted regulatory pathways (*Westra et al., 2013*; *Fagny et al., 2017*). Therefore, for a better disease understanding and to take eQTLs one step further, one has to look beyond the disruption of individual genes and determine how the interaction of genes changes based on cell type (*Battle et al., 2014*; *GTEx Consortium, 2017*; *Westra et al., 2015*; *Knowles et al., 2017*), environment (*Favé et al., 2018*; *van der Wijst et al., 2018b*) and an individual's genetic makeup (*Zhernakova et al., 2017*; *van der Wijst et al., 2018a*). The sc-eQTLGen consortium will do so

by reconstructing personalized, cell type-specific GRNs (*La Manno et al., 2018*; *Figure 3*). The unique features of scRNA-seq data, among which the inference of pseudotime and RNA velocity (i.e. the ratio between spliced and unspliced mRNA that allows prediction of the future state of a cell) (*Qiu, 2018*), enable learning the directionality of network connections (*Fiers et al., 2018*). We expect that such personalized GRNs will help explain for example differences in interindividual drug responses, and thereby, will aid in precision medicine in the future.

Reconstruction of GRNs from single cell data reviewed in *Raj et al. (2006)* is complicated by the sparsity of the data as a consequence of the stochasticity underlying gene expression (*Chen and Mar, 2018*) and dropouts, i.e. genes that are not detected in some cells as a consequence of technical limitations (*Jackson et al., 2019*). This sparsity leads to lower correlation estimates that obscure the identification of true edges in the GRNs. Several solutions have been developed to overcome this problem, including the implementation of prior information (*Aibar et al., 2017*; *Andrews and Hemberg, 2018*), gene expression imputation (*Aibar et al., 2017*; *Iacono et al., 2019*) and usage of alternative measurements of correlation (*Skinnider et al., 2019*; *Budden et al., 2014*).

Firstly, prior information encoded in the DNA sequence can be used to overcome these complications (*Angermueller et al., 2017*; *Angelini and Costa, 2014*). Such priors on regulatory interactions can be derived from, for example, ChIP-seq data (*Miraldi et al., 2019*), ATAC-seq data (*Qin et al., 2014*), spatial information (*Burgess, 2019*; *Maynard et al., 2020*) or from perturbation experiments (*Gate, 2019*; *Aibar et al., 2017*). Implementation of such priors was shown to improve bulk GRN reconstruction (*Qin et al., 2014*; *Ghanbari et al., 2015*; *Azizi et al., 2018*), and similarly, it is expected to also improve GRNs reconstructed from single-cell data (*Aibar et al., 2017*; *Andrews and*

*Hemberg, 2018*). However, caution is warranted when using this information, as their effect on GRN reconstruction depends on the quality of these data priors (*Siahpirani and Roy, 2017*; *Simpson, 1951*) and priors derived from bulk data may not hold true at the single-cell level (*Buenrostro et al., 2015*). Recent technological advances enable studying chromatin accessibility (*Lareau et al., 2019*; *Hayashi et al., 2018*) and expression of enhancers RNAs (*Kouno et al., 2019*; *Lin et al., 2016*) at the single cell level, which will make it possible to implement single-cell derived priors in GRN reconstruction in the future, though these quantifications come with their own limitations and challenges.

Secondly, gene expression imputation may be used to restore the underlying correlation structure. However, current gene expression imputation methods become more unreliable as the dropout rates increase (*Iacono et al., 2019*; *Skinnider et al., 2019*). After gene expression imputation, more network edges are identified, but with a higher chance of detecting false positives (*Aibar et al., 2017*; *Iacono et al., 2019*). Nevertheless, by combining prior information with imputation, GRN reconstruction can be improved both in the bulk (*Qin et al., 2014*) and single cell setting (*Aibar et al., 2017*). For example, one can replace transcription factor expression with inferred transcription factor activities based on the collective expression patterns of their target genes or take advantage of cross-omics relationships (*Stoeckius et al., 2017*).

Finally, alternative correlation measures are being explored to overcome the complications associated with data sparsity, including

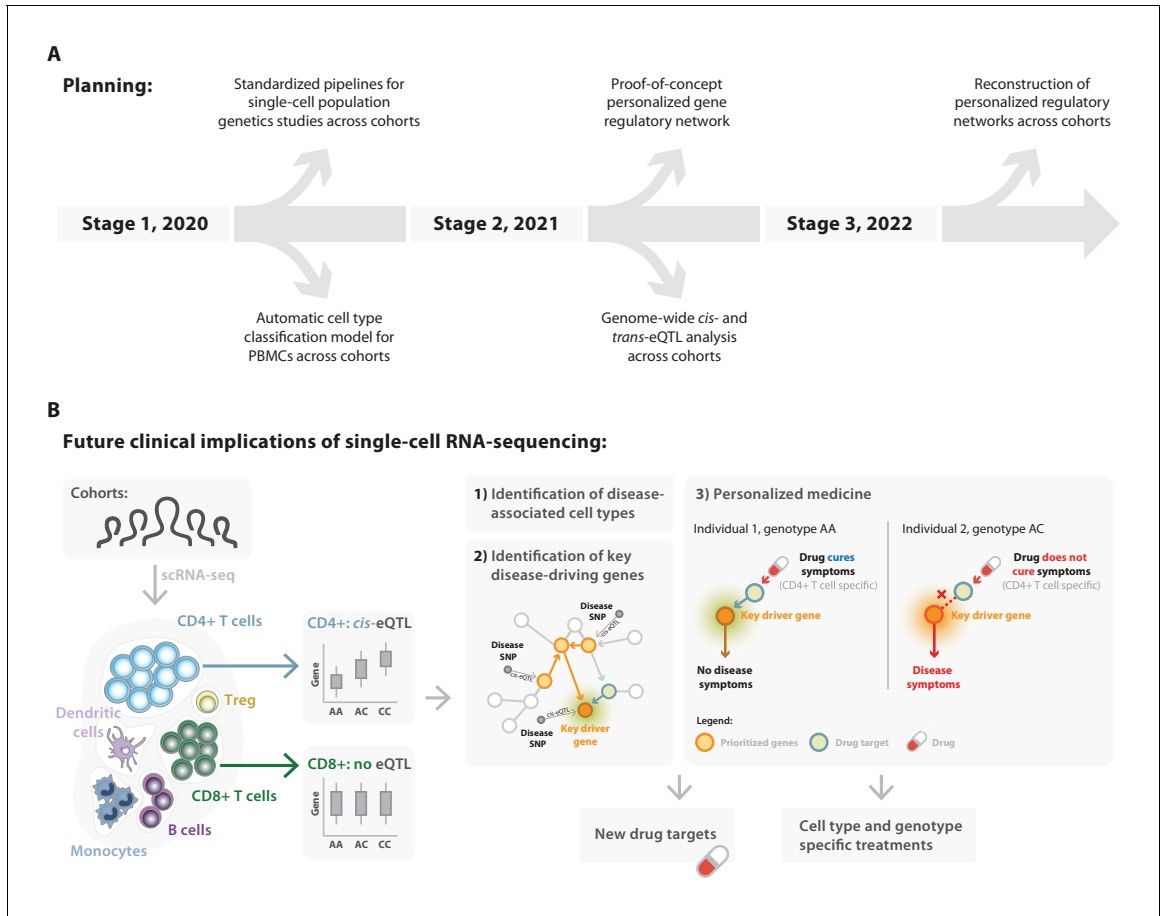

**Figure 5.** Deliverables of the single-cell eQTLGen consortium in relation to their future clinical implications. (**A**) In the coming three years the sc-eQTLGen consortium aims to deliver the following: i) standardized pipelines and guidelines for single-cell population genetics studies (2020); ii) cell type classification models for PBMCs (2020); iii) summary statistics of *cis*- and *trans*-eQTLs, co-expression QTLs, cell count QTLs and variance QTLs (2021); iv) reconstruction of personalized, cell type-specific gene regulatory networks (2022). (**B**) These efforts of the consortium will lead to the (**1**) identification of disease-associated cell types and (**2**) key disease-driving genes, which together will aid (**3**) the implementation of personalized medicine and the development of new therapeutics that take all this information into account (cell type- and genotype-specific treatments). Panel B2 is adapted from *van der Wijst et al. (2018b)*.

measures of proportionality (*Budden et al., 2014*) and by calculating the correlations on measures other than the normalized expression counts (*Skinnider et al., 2019*). For example, Z-scores of the gene expression distributions of highly similar cells have been used to calculate the co-expression relationships. This approach could reveal the true correlation structure that was otherwise hidden by technical artifacts (*Skinnider et al., 2019*). In addition to these computational tools, technological advances, such as single-cell multi-omics approaches (*Cao et al., 2018*; *Moignard et al., 2015*) and improved experimental protocols, are expected to alleviate these complications. Moreover, being able to assess multiple layers of information within the same cell, e.g. chromatin accessibility, DNA methylation, gene and protein expression, opens unique opportunities for developing new methodology for GRN reconstruction and validation. Altogether, this will further improve the accuracy of GRNs reconstructed using single-cell data in the future.

The incorporation of dynamic information extracted from time series or pseudotime (*Ocone et al., 2015*; *Pratapa et al., 2019*) is another promising avenue to further improve single-cell GRN reconstruction. However, not all datasets are equally well suited to identify temporal trajectories. For example, PBMCs are usually in steady state, and only after stimulation such trajectories would appear.

Summarized, the ideal GRN reconstruction tool can efficiently manage large amounts of single-cell data, incorporate prior information, model non-linear relationships and take dynamic information into account. Early benchmark studies, performed for a limited number of methods on rather small datasets (*Jackson et al., 2019*) or on simulated data (*Lukowski et al., 2017*) show that current tools usually only work well in specific situations. As such, there is a clear need for the development of all-round tools that work well in every situation.

## sc-eQTLGen: a federated single-cell eQTL meta-analysis consortium

Combining data of numerous groups increases the resolution and power by which downstream analyses, such as eQTL identification and personalized GRN reconstruction, can be performed. Ideally, all scRNA-seq datasets should be jointly analyzed at one centralized location.

This is particularly helpful to align each group's approaches for preprocessing, quality control (QC) and cell type classification. However, it also eases for instance benchmarking different statistical and computational methods. While this concept of 'bringing the data to the algorithm' is preferred from an analytical perspective, it is usually very difficult to do so when handling privacy-sensitive scRNA-seq and genotype data from human individuals (*Lloyd-Jones et al., 2017*; *Yengo et al., 2018*).

To overcome this, a federated approach could be used instead, which has the aim of 'bringing the algorithm to the data': each participating cohort will run the analyses themselves (adhering to predefined criteria for preprocessing and QC), and will only share summary statistics that are not privacy-sensitive. Finally, one site takes responsibility for performing the overall meta-analysis using these provided summary statistics. For genome-wide association studies this is a common strategy (*Xue et al., 2018*; *Luecken and Theis, 2019*), and for eQTL studies this procedure has been shown to be effective as well (*Võsa, 2018*; *Westra et al., 2013*). In the following sections we will expand on all steps that have to be taken and what considerations should be made when conducting such a federated approach for single-cell population genetics studies (*Figure 4*).

### Preprocessing, quality control

The first challenge of federated analyses is the need to have a standardized protocol on how each group should perform their analyses. While such a protocol helps to ensure reproducibility of the data analysis, it requires that all methods and tools used have been rigorously tested before. For scRNA-seq data such protocols are still under development, while in other fields such as that of genome-wide association studies, standardized protocols have been available for years.

Several initiatives are now being undertaken to define best practices in the scRNA-seq field (*Tian et al., 2019*). For example, Tian et al. have compared 3913 combinations of different scRNA-seq data analysis pipelines to define best practices in the field (*Price et al., 2006*). Such initiatives could provide the basis for defining the optimal preprocessing, QC and cell type classification steps for our consortium. Additionally, in population-based scRNA-seq studies special attention is required to account for ethnic variation and population stratification (see *Box 1*; *Zeng and Gibson, 2019*; *Zhou and*

*Stephens, 2012*). In the event of presence of relatedness in a given cohort, a genetic relatedness matrix will be included in a mixed model to account for the effect, such as in *Zhou and Stephens (2012)*; *Abdelaal et al. (2019)*. Adjustments of cohort-level genetic differences will be made in the framework of meta-analysis using summary statistics of the individual cohorts. Once all protocols are established, we can harmonize the preprocessing steps across all groups in the consortium, such as the genome build to use, alignment tool and sample demultiplexing strategy. Due to the cohort-specific characteristics of each dataset, the QC steps cannot be harmonized to the same extent as the preprocessing. Nevertheless, the parameters used for QC can be coordinated across all groups, such as the cutoffs for number of detected genes per cell and mitochondrial fraction. Both the preprocessing and the QC do not require exchanges of data and can be performed independently.

### Cell type classification

To facilitate the eQTL meta-analysis, we need to ensure that the cell type annotations are consistent across the different cohorts. To ensure reproducibility of annotations across the different cohorts, we will employ a classification scheme to identify canonical cell types in each cohort separately. Performing cell type labeling using classification models does not only increase the reproducibility, but also constitutes a privacy-safe way of annotating cell types that does not require the sharing of raw or processed gene expression data.

Reference datasets with labeled cells, such as those available from the Immune Cell Atlas (http://immunecellatlas.net/) will be used to train a classifier for automatic cell type classification in each cohort. Our recent comparisons of single cell classification methods showed that simple linear models can yield good results (*Köhler et al., 2019*; *Wolf et al., 2019*). Despite the wide availability of reference datasets, we expect that some cohorts will contain novel unknown cell types or states that cannot be identified using the trained classifier. For this, we will use a classification scheme with a rejection option that can flag unknown cells whenever the confidence in cell type assignment is low (*Köhler et al., 2019*). The rejected cells can then be manually annotated based on marker gene expression.

To capitalize on the large number of cells and individuals to be profiled in each cohort, an unsupervised clustering approach will be used to analyze the count matrix of each cohort, in parallel to the supervised approach described earlier. This unsupervised approach will serve two purposes: (1) it will help annotate unassigned cells by the classifier, and (2) it will allow refining the resolution at which cells are annotated. Varying levels of granularity of the clustering may reveal cell types, as well as particular cell states or subtypes. This level of granularity required to separate particular cell states is not known a priori. Therefore, novel unbiased approaches such as partition based graph abstraction (*Baran, 2018*) or metacells, i.e. disjoint, homogenous and highly compact groups of cells that each exhibit only sampling variance (*Saelens et al., 2019*), provide a framework to reconcile discrete states at different levels of granularity with continuous cell states. These novel annotations can feed back into an iterative online learning approach of supervised classification models, where we could refine cell type prediction models on the available datasets. Once new datasets become available within the consortium these can be annotated based on current models and updated labels can be used in the next round of training. An important consideration here is to preserve the hierarchy of cell annotations, so that if new annotations are added to the classifier, they are subclasses of existing classes. In this way, any downstream analysis based on older annotations remains valid at the older level of granularity. This would yield a coherent approach of labeling over time as the dataset grows. For inference of continuous cell states, we require data integration across multiple centers, as this would ensure the usage of a similar pseudotime scale between individuals. Currently, ordering cells along pseudotime is challenging and best practices are being evaluated (*Price et al., 2006*; *Roshchupkin et al., 2016*).

Ultimately, integrating all expression data in a privacy-preserving manner, i.e. as gene expression matrices, will produce a dataset with unprecedented numbers of cells. Such a dataset allows discovery of novel rare cell types or states using clustering approaches as described above. This valuable dataset will then be shared with the community through platforms like the HCA data portal.

### eQTL and co-expression QTL analysis

After cell type assignment, annotated gene expression matrices can be returned to each of the cohorts. Each cohort will then map genome-wide cell type-specific *cis-* and *trans-*eQTLs by

combining the cell type-specific gene expression matrices with the privacy-sensitive bulk-assessed genotype information using appropriate statistical models. The resulting summary-statistics can then be safely shared without privacy-issues.

One challenge with federated eQTL analyses is that the amount of summary statistics that need to be shared is substantial. For instance, when assuming there are 10 cohorts and for each of these cohorts cells have been assigned to 10 major cell types, a genome-wide *trans*-eQTL analysis (testing the effect of 10,000,000 common SNPs on 20,000 protein coding genes for each of the 10 cell types), where only the correlation for a SNP-gene combination is stored as a 64 bit double value, would require each cohort to exchange 10,000,0000 $\times$ 20,000$\times$10 x eight bytes = 146 terabytes of data. To overcome this problem, several frameworks have recently been proposed that take advantage of the fact that these summary statistics matrices reflect the product of a normalized genotype matrix and a normalized gene expression matrix. For instance, the HASE framework (*Silvester et al., 2018*) recodes genotype and phenotype (i.e. gene expression) data, along with a covariate matrix, in such a way that privacy is ensured and only those matrices, making up only a few gigabytes of data, need to be exchanged.

While protocols exist that explain how eQTL data needs to be processed, harmonized and QCed to perform a federated eQTL analysis (e.g. eQTLGen used the eQTLMappingPipeline [*Võsa, 2018*]), not all steps can be completed immediately: for instance, to identify effects of polygenic risk scores on gene expression levels (ePRS), gene expression data first needs to be corrected for *cis*-eQTL effects (*Võsa, 2018*). Therefore, the full *cis*-eQTL meta-analysis has to precede calculations of ePRSs. Such iterations take considerable time and are also inconvenient, since it requires a lot of coordination with each of the participating cohorts. For sc-eQTL-Gen we will first conduct a federated, cell type-specific *cis*- and *trans*-eQTL analysis. After this is completed, we will proceed with a co-expression QTL (co-eQTL) analysis. This analysis will be limited to a predefined set of genes or SNPs, such as the SNP-gene combinations extracted from the identified *cis*- and *trans*-eQTLs or the SNPs located within open chromatin regions that show high interindividual variability, as otherwise trillions of statistical tests have to be conducted (e.g. in *van der Wijst et al., 2018a*: 7975 variable genes * 7975 variable genes * 4,027,501 SNPs (MAF $\geq$0.1) = 256,151,580,788,125 tests).

Finally, all these results will be combined to reconstruct personalized, cell type-specific GRNs. This multi-step approach will require us to go back and forth between the different cohorts at least twice. Therefore, easy-to-use analysis scripts that can be run efficiently on different high-performance cluster infrastructures are essential to limit the amount of hands-on time.

### Gene regulatory network reconstruction

Finally, the scRNA-seq data will be used to reconstruct GRNs. Two strategies will be explored in the context of sc-eQTLGen. The first approach makes use of the large number of bulk RNA-seq datasets for specific cell types that are available in public RNA-seq repositories (*Leinonen et al., 2011*; *Langfelder and Horvath, 2008*). Using this publicly available bulk RNA-seq data, reference co-expression networks will be constructed using cell type-specific data. Subsequently, scRNA-seq data will be used to implement directionality and specify the connections in the network that are affected by specific contexts (*La Manno et al., 2018*). The second approach will directly use scRNA-seq data to build cell type-specific GRNs, thereby enabling to immediately take the context-specificity into account. However, the number of genes that can confidently be taken into account by this second approach may be lower due to the sparsity of scRNA-seq data. For both strategies, we will make use of prior information (e.g. ATAC-seq data [*Qin et al., 2014*], TF binding information), dynamic information (e.g. information extracted from time series data (*Ocone et al., 2015*), pseudotime (*Pratapa et al., 2019*) in combination with RNA velocity (*Qiu, 2018*; *Fiers et al., 2018*) and experimental validation (e.g. perturbation experiments [*Gate, 2019*; *Aibar et al., 2017*]) to go from a co-expression to a gene regulatory network. Before implementation, the additional benefit of using such information, extracted from either bulk or single-cell data (*Aibar et al., 2017*; *Andrews and Hemberg, 2018*), and using gene expression imputation (*Aibar et al., 2017*; *Iacono et al., 2019*) will be assessed. We expect that the optimal strategy will depend on the amount of available bulk data and prior information that is available for a particular cell type. We will extract this prior information from existing large-scale efforts, such as ENCODE (*Wang et al., 2018*) and BLUE-PRINT (*Chen et al., 2016*). Additionally, we will make use of single-cell information beyond gene expression levels that is or will be collected

within subsets of cohorts within the consortium, including information on chromatin accessibility (*Lareau et al., 2019*; *Hayashi et al., 2018*) and expression of enhancers RNAs (*Kouno et al., 2019*; *Lin et al., 2016*).

Additionally, recent advances have made it possible to measure multi-omics data from the very same cell (*Moignard et al., 2015*; *Sverchkov and Craven, 2017*). However, current approaches are very time- and cost-consuming, and therefore limited to only a few hundred cells. As such, currently, this type of single-cell multi-omics data is of limited use for reconstructing personalized GRNs. Nevertheless, as single-cell multi-omics approaches mature, this combined information of gene expression and additional data layers has the potential to improve GRN inference beyond correlating separate omics layers and allows for direct measurements instead.

Once reconstructed, these GRNs can be used to determine how for example, genetic differences or disease status change the architecture of the network. These networks consist of nodes, representing genes, that are connected through edges, representing the relationship between genes. The context-specific changes in the network can be identified on different levels, such as on the level of individual edges or nodes, topological properties of individual nodes, such as their connectivity (degree) or module membership (*GTEx Consortium, 2015*), subnetwork properties, such as the existence and size of modules, or global topological properties, such as degree distribution (*Figure 3*). Comparing topological features such as node degree to genotypes may identify polymorphisms altering the function of master regulators (highly connected 'hub' genes). Interestingly, implementation of network information was shown to be complementary to the identification of eQTLs; using this network information, novel SNPs were identified that could not be identified through single- or multi-tissue eQTL analyses of GTEx (*Clark et al., 2018*). This clearly shows the complementarity of both eQTL and network-based analyses for understanding the impact of genetic variation.

Ultimately, CRISPR perturbations will be coupled to scRNA-seq to validate or improve reconstructed GRNs. To optimize the number of perturbations required for extracting the most useful information from such experiments, an iterative approach will be taken that feeds back the experimental data to the GRN. This approach will make use of active machine learning to select those perturbations that are required to further improve the model (*Ud-Dean and Gunawan, 2016*; *Aguet, 2019*). These well validated, personalized and context-specific GRNs will provide us with a better understanding of disease and can be the starting point of applying this knowledge for precision medicine in the future.

## Future clinical implications

The goal of the sc-eQTLGen consortium is to identify how genetic and environmental factors interact to affect gene expression in the context of both health and disease. With ever increasing sample sizes, eQTLs have now been detected for almost every gene (*GTEx Consortium, 2017*; *Võsa, 2018*). It is likely this will become even more pronounced through our initiative in which we will study many different cell types and contexts, and pose the question to what extent extensive eQTL maps will help to better understand disease. For *cis*-eQTLs this will not be straightforward: although it is known that disease-associated SNPs are enriched for showing *cis*-eQTL effects, this enrichment is quite modest (*Zhernakova et al., 2017*; *Liu et al., 2019*). It is therefore not sufficient to simply look up and catalogue which disease-associated SNPs show which cell type- and context-specific eQTL effects, since this can lead to incorrect inferences on the likely causal gene(s) per locus (*Giambartolomei et al., 2014*). To partly overcome this, several colocalization and Mendelian randomization approaches have been published that help to better infer likely causal genes (*Gusev et al., 2016*; *Porcu et al., 2019*; *Mancuso et al., 2019*; *Kemp, 2019*). Once these methods are able to account for multiple cell type- and context-dependent, causal regulatory variants per locus, we expect increased statistical power to prioritize causal genes. Additionally, we envision that such methods in conjunction with our cell type- and context-specific eQTL maps will help to determine which genetic variants have pleiotropic effects, affecting the expression levels of several genes in multiple cell types and conditions.

Nevertheless, we expect that most statistical power to pinpoint causal genes will be gained through the other goals of the sc-eQTLGen consortium: the reconstruction of cell type-specific gene regulatory networks (expected by the end of 2022), the mapping of cell type-specific *trans*-eQTLs and co-expression QTLs (expected by the end of 2021). These efforts will enable us to

ascertain how the prioritized *cis*-eQTL genes (expected by the end of 2021) work together. Moreover, it permits us to study the effect of all disease-associated SNPs of a particular disease in a gene network structure, which helps prioritizing the key disrupted genes and pathways in that disease. In line with recent findings of the eQTLGen consortium that applied eQTL analysis in 31,684 bulk samples, we expect that the (majority of) causal genes within disease-associated loci will converge onto only a few key pathways per disease.

One strategy to identify those key driver genes is to consider all associated variants for a specific disease jointly, and ascertain whether most of these variants show (small) downstream effects on an overlapping set of downstream genes. We recently showed proof-of-concept in eQTLGen that this holds for independent systemic lupus erythematosus (SLE)-associated SNPs: many of these variants show downstream *trans*-eQTL effects on genes involved in the type I interferon pathway (*Võsa, 2018*), indicating an important role for this pathway in SLE development. Recently, success has been reported of a type I interferon-targeted therapy in SLE patients (*Astle et al., 2016*), highlighting the value of using *trans*-eQTLs for identifying key genes and pathways that are amenable for pharmaceutical intervention. We expect that our single-cell eQTL initiative will aid such analyses substantially: by performing large-scale eQTL mapping in specific cell types that are in a specific cellular state or are exposed to a particular stimulus, we will be able to more accurately determine where and when these downstream effects manifest. Moreover, single-cell studies will also help to overcome the problem associated with cell type composition differences across individuals in bulk-based eQTL studies: many variants exist that affect the proportion of specific cell types that for instance circulate in blood (*Vieira Braga et al., 2019*). If this is not fully accounted for, *trans*-eQTLs will be identified in genes that are specifically expressed in such cell types in bulk analyses. Single-cell studies allow us to distinguish between effects of genetic variants on cell type composition and effects on intracellular gene expression levels. Therefore, we expect scRNA-seq data will be vital to gain insight into the downstream consequences of disease-associated genetic variants, and to identify the key pathways and genes that drive disease.

Altogether, we expect these approaches will provide us with the information required to reveal new targets for disease prevention and treatment (*Figure 5*). For example, a novel subset of tissue-resident memory T cells has recently been identified in the setting of asthma using scRNA-seq (*Schork, 2015*). This study also showed that mostly T helper 2 cells are dominating the cell-cell interactions in the asthmatic airway wall, whereas in healthy controls mostly epithelial and mesenchymal cell types are communicating with each other. Integration of the gene expression of this asthma-associated cell type with asthma-associated genetic risk variants would further increase our understanding of the disease and such knowledge would greatly accelerate the development of personalized/precision treatments in the future. It is this information about how genes interact differently between individuals as a function of their genetic predisposition that will be obtained through the results of our consortium (*Figure 5*). One of the major benefits of such personalized treatments is in prescribing the correct drug based on the individual (mechanism that underlies) susceptibility to disease. Currently only between 4% and 25% of the people respond to commonly prescribed drugs (*Mukherjee and Topol, 2002*), showing the need to better predict drug responsiveness and thereby avoid unnecessary exposure to side-effects.

This high interindividual variability in drug response is a consequence of genetic and environmental exposure differences between individuals, which can result in differences in drug metabolism, absorption and excretion (pharmacodynamics) (*Johnson et al., 2012*). For example, a variant in the *CYP2C19* gene changes the response to the anti-blood clotting drug clopidogrel. The *CYP2C19* gene encodes for an enzyme in the bioactivation of the drug. *CYP2C19* poor metabolizers were shown to exhibit higher cardiovascular event rates after acute coronary syndrome, or percutaneous coronary intervention, as compared to patients with normal *CYP2C19* function (*Tigchelaar et al., 2015*).

While previous efforts have mainly focused on pharmacodynamic variation, recent single-cell analyses have revealed that gene-gene interactions can also be changed by genetic (*van der Wijst et al., 2018a*) and environmental variation (*Cuomo et al., 2020*). For example, two closely related SNPs (linkage disequilibrium $R^2$ = 0.92) affected both gene-gene interactions (*RPS26* and *RPL21*) (*van der Wijst et al., 2018a*) and gene-environment interactions (*RPS26* and the respiratory status of the cell) (*Cuomo et al.,*

*2020*). This shows that gene regulatory network changes may underlie part of the interindividual variation in drug responsiveness. However, such effects have never been studied in detail before and the extent to which such interactions affect drug responsiveness are unknown. The sc-eQTL-Gen consortium is able to study both how gene-gene interactions and gene-environment interactions are affected by genetic variation, giving insight into where and when they occur. Importantly, the applied methodologies will be easily transferable to single-cell data that is collected in other cell types and disease context through other large-scale efforts (*Regev et al., 2017*) (https://lifetime-fetflagship.eu). Moreover, several partners within our consortium have generated scRNA-seq data in cohorts with extensive information on individuals' health records and drug usage (e.g. the Lifelines Deep cohort [*Heaton, 2019*] and the OneK1K cohort). With such information, we will be able to validate the link between changes in the gene regulatory network and the drug responsiveness of an individual. This allows us to determine the predictive value of gene networks for determining responsiveness of specific drugs and the applicability of such networks in precision medicine.

As such, the sc-eQTLGen consortium will not only increase our basic knowledge about the contribution of genetics in gene expression and its regulation, but will also be a valuable resource for drug target identification and validation. To increase the impact of this work, all code, guidelines and summary statistics (including all non-significant results) will become freely available to the community through the sc-eQTLGen website (https://eqtlgen.org/single-cell.html). For any additional information, please visit the contact page (https://eqtlgen.org/contact.html).

**MGP van der Wijst** is in the Department of Genetics, Oncode Institute, University of Groningen, University Medical Center Groningen, Groningen, Netherlands
m.g.p.van.der.wijst@umcg.nl
https://orcid.org/0000-0003-1520-3970

**DH de Vries** is in the Department of Genetics, Oncode Institute, University of Groningen, University Medical Center Groningen, Groningen, Netherlands.

**HE Groot** is in the Department of Cardiology, University of Groningen, University Medical Center Groningen, Groningen, Netherlands
https://orcid.org/0000-0002-8265-3085

**G Trynka** is at the Wellcome Sanger Institute and Open Targets, Wellcome Genome Campus, Hinxton, United Kingdom
https://orcid.org/0000-0002-6955-9529

**CC Hon** is at the RIKEN Center for Integrative Medical Sciences, Yokohama, Japan

**MJ Bonder** is in the Genome Biology Unit, European Molecular Biology Laboratory, Heidelberg, and the Division of Computational Genomics and Systems Genetics, German Cancer Research Center (DKFZ), Heidelberg, Germany
https://orcid.org/0000-0002-8431-3180

**O Stegle** is in the Genome Biology Unit, European Molecular Biology Laboratory, Heidelberg, and the Division of Computational Genomics and Systems Genetics, German Cancer Research Center (DKFZ), Heidelberg, Germany

**MC Nawijn** is in the Department of Pathology and Medical Biology, GRIAC Research Institute, University of Groningen, University Medical Center Groningen, Groningen, Netherlands
https://orcid.org/0000-0003-3372-6521

**Y Idaghdour** is in the Program in Biology, Public Health Research Center, New York University Abu Dhabi, Abu Dhabi, United Arab Emirates
https://orcid.org/0000-0002-2768-9376

**P van der Harst** is in the Department of Cardiology, University of Groningen, University Medical Center Groningen, Groningen, Netherlands
https://orcid.org/0000-0002-2713-686X

**CJ Ye** is in the Institute for Human Genetics, the Bakar Computational Health Sciences Institute, the Bakar ImmunoX Initiative, the Department of Medicine, the Department of Bioengineering and Therapeutic Sciences, the Department of Epidemiology and Biostatistics, and the Chan Zuckerberg Biohub, University of California San Francisco, San Francisco, CA, United States

**J Powell** is in the Garvan-Weizmann Centre for Cellular Genomics, Garvan Institute, UNSW Cellular Genomics Futures Institute, University of New South Wales, Sydney, Australia

**FJ Theis** in the Institute of Computational Biology, Helmholtz Zentrum München, Neuherberg, and the Department of Mathematics, Technical University of Munich, Garching bei München, Germany
https://orcid.org/0000-0002-2419-1943

**A Mahfouz** is in the Leiden Computational Biology Center, Leiden University Medical Center, Leiden, and the Delft Bioinformatics Lab, Delft University of Technology, Delft, Netherlands
https://orcid.org/0000-0001-8601-2149

**M Heinig** in in the Institute of Computational Biology, Helmholtz Zentrum München, Neuherberg, and the Department of Informatics, Technical University of Munich, Garching bei München, Germany
https://orcid.org/0000-0002-5612-1720

**L Franke** is in the Department of Genetics, Oncode Institute, University of Groningen, University Medical Center Groningen, Groningen, Netherlands
https://orcid.org/0000-0002-5159-8802

*Author contributions:* MGP van der Wijst, Conceptualization, Visualization, Project administration, Writing - original draft; DH de Vries, J Powell, A Mahfouz, M Heinig, Conceptualization, Visualization, Writing - original draft; HE Groot, Visualization, Writing - original draft; G Trynka, Visualization; CC Hon, Writing - original draft; MJ Bonder, O Stegle, P van der Harst, Writing - review and editing; MC Nawijn, Y Idaghdour, Writing - original draft, Writing - review and editing; CJ Ye, Conceptualization, Writing - original draft; FJ Theis, Conceptualization; L Franke, Conceptualization, Writing - review and editing

*Competing interests:* The authors declare that no competing interests exist.

## Funding

| Funder | Grant reference number | Author |
| --- | --- | --- |
| Nederlandse Organisatie voor Wetenschappelijk Onderzoek | NWO-Veni 192.029 | MGP van der Wijst |
| Nederlandse Organisatie voor Wetenschappelijk Onderzoek | ZonMW-VIDI 917.14.374 | L Franke |
| European Research Council | ERC Starting grant Immrisk 637640 | L Franke |
| Oncode Institute | | L Franke |
| National Health and Medical Research Council | Investigator grant 1175781 | J Powell |

The funders had no role in study design, data collection and interpretation, or the decision to submit the work for publication.

### Decision letter and Author response

Decision letter https://doi.org/10.7554/eLife.52155.sa1
Author response https://doi.org/10.7554/eLife.52155.sa2

## Additional files

### Data availability

Not applicable.

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
