## [Decision Letter]

Thank you for submitting your article "The single-cell eQTLGen Consortium" for consideration by *eLife*. Your article has been reviewed by three peer reviewers, and the evaluation has been overseen by two editors from the *eLife* Features Team (Helena Pérez Valle and Peter Rodgers). The following individuals involved in review of your submission have agreed to reveal their identity: Stephen B. Montgomery (Reviewer #1); Maud Fagny (Reviewer #2).

The reviewers have discussed the reviews with one another and the editors have drafted this decision to help you prepare a revised submission.

The manuscript introduces the single-cell eQTLGen Consortium, which aims to assess the effect of genetic variants on gene expression at the cell-type level, and describes the goals of the consortium and its plans for data analysis. The authors propose the development of standardized guidelines and pipelines to perform eQTL analyses, as well as a roadmap to perform analyses while preserving anonymity. They highlight the expected results of the Consortium, including a better understanding of the molecular bases of complex diseases and of the cell type involved, and potential clinical applications.

The manuscript is clearly written and interesting, and it outlines a sound an applicable protocol to analyse many cohorts while preserving data privacy. However, it would benefit from addressing a number of issues in greater detail - see below.

Essential revisions:

1. The Consortium does not provide detail of the study design considerations for data producers in any specifics. I.e. how data should be processed, how many cells/individual, how many reads. Minimum number of individuals. Inclusion/exclusion criteria. Further, what type of genotyping will be required for individuals. For example, the authors mention that cells from different individuals can be mixed together and "multiplexed" to reduce cost and avoid confounding, but it would be very useful for the authors to show the mapping power increase obtained from multiplexing. Analyses like this may help researchers decide on their preferred collection design which would allow a better harmonization of data generated from outside the consortium.

2. The analyses proposed are exciting but the specifics of how they will be run are vague. It would be helpful to catalogue existing tools and identify where new tools are needed, highlighting where the code/algorithms will eventually be found.

3. Authors mention gene regulatory networks, when they really plan to study gene co-expression network. "Regulatory" suggests a causal relationship between 2 nodes, while co-expression only relies on correlations. While similar changes in expression levels among cells might suggest a co-regulation, no inference can be made about a regulatory relationship between genes in absence of complementary information such as TF bindings. Some approaches are able to build regulatory networks from expression data, with the addition of prior information (see Sonawane et al., Network Medicine in the age of biomedical big data. 2019. Frontiers in Genetics. doi: 10.3389/fgene.2019.00294).

4. scRNA-seq data has lower power for eQTL mapping than bulk RNA-seq when matched for sample size. It would be informative for the readers and community to get a better sense of the number of eQTLs that we would expect to map based on individual sample size, number of cells captured by experiment, cell-type proportion in PBMC, etc...

5. Many data are mentioned (genomic data, scRNA-seq, scATAC-seq, sc-protein level...), but it is not always clear which ones will be generated, which ones may be generated, and which ones are already existing datasets. Maybe a figure would help?

6. There is limited mention of potential ASE-based or splicing analyses.

7. There is limited mention of how multi-omics from single cell data may improve GRN or other analyses. There are multiple studies that have now obtained different data modalities from the same cells.

8. I would expect some discussion of spatial transcriptomics and its potential.

9. How does the consortium and its work relate to/differ from the following project?

https://chanzuckerberg.com/science/programs-resources/humancellatlas/seednetworks/human-immune-variation-across-genetic-backgrounds-gender-and-time/

10. Please outline the deliverables proposed for the consortium (including a timeline for when they will be available).

11. Aspects of future data sharing and accessibility are essential to address.

12. Please explain how new individuals can become members of the consortium.

13. Please explain the consortium will be funded.

---

## [Author Response]

[We repeat the reviewers’ points here in bold text, and include our replies point by point, as well as a description of the changes made, in plain text].

Reply to reviewer comments:

We would like to thank the reviewers and editors for their highly constructive comments and suggestions, which have enabled us to improve our manuscript substantially. We believe that we have now adequately addressed each reviewer’s comments. In short, we have added additional information about data sharing and how to become part of this effort. Moreover, we have further improved the figures and resolved some confusions. Finally, we have added an overview of the deliverables and the associated timelines in Figure 5. This will provide the reader with insights into when to expect standardized pipelines and guidelines for designing single-cell population genetics studies, and when to expect our first results.

Essential revisions:1. The Consortium does not provide detail of the study design considerations for data producers in any specifics. I.e. how data should be processed, how many cells/individual, how many reads. Minimum number of individuals. Inclusion/exclusion criteria.

We completely agree that any detail of study design considerations would greatly benefit groups that are interested in conducting single-cell population genetics studies and joining the consortium in the future. Unfortunately, we do not have all the answers yet.

In Box 1 we do already mention some more general guidelines related to sample multiplexing:

*“*First of all, the genetic information that is available for each of the individuals in such cohorts can be used to demultiplex pools of multiple individuals within the same sample.This approach allows to properly randomize experiments, while also significantly reducing cost and confounding effects^23, 107^. This genetic information can either be efficiently generated using genotype arrays^108^ in combination with imputation-based approaches^109^, or extracted from the scRNA-seq data itself^107, 110^. Within the consortium all reads will be aligned to the *GRCh38/hg38* reference genome and genotypes will be imputed using the Haplotype Reference Consortium reference panel^111^. The basic principle behind genetic multiplexing is that enough transcripts harboring SNPs are expressed and detected in each single cell such that cells can be accurately assigned to the donor of origin. Furthermore, as the number of multiplexed individuals increases, the probability that a droplet harbors multiple cells from different individuals increases, thus allowing the detection of multiplets using genetic information. This enables the overloading of cells into standard droplet-based workflows and overall reduction of cost per cell up to about 10-fold (https://satijalab.org/costpercell). As the cost of sequencing and the background multiplet rate reduce, the benefits of multiplexing increase. We anticipate that future workflows will allow for even higher throughput.”

… and the number of cells and individuals required for identifying eQTLs, and how to most cost-efficiently design such studies for these purposes:

“Finally, studying genetic variation at the single-cell level adds some extra requirements for the number of cells per individual and the number of individuals to be included in the study. The number of cells per individual will mainly define for which cell types in a heterogeneous sample such as PBMCs eQTL and co-eQTL analyses can be performed. In contrast, the number of individuals will mainly define the number of genetic variants for which effects on gene expression can be confidently assessed. A recent analysis showed that, with a fixed budget, the optimal power for detecting cell type-specific eQTLs is obtained when the number of reads is spread across many individuals^118^. Even though a lower sequencing depth per cell results in a lower accuracy of estimating cell type-specific gene expression levels, many more individuals and cells per individual can be included for the same budget. As a result, the optimal experimental design with a fixed budget provides up to three times more power than a design based on the recommended sequencing depth of 50,000 reads per cell (for 10X Genomics scRNA-seq). In contrast, for co-eQTL analysis there is a different trade-off between sequencing depth, number of individuals and number of reads per cell; while for eQTL analysis gene expression levels among cells of the same cell type can be averaged, for co-eQTL analysis you cannot as this would prohibit you from calculating a gene-gene correlation per individual. Therefore, for co-eQTLs the sequencing depth will be a major limiting factor that determines the number of genes for which you can confidently calculate gene-gene correlations. Altogether, depending on the goal of your study, the optimal balance between sequencing depth and number of individuals and cells per individual will be different.”

However, the consortium aims to have more detailed guidelines ready by the summer of 2020. Currently, the members of the consortium are working on defining the minimum requirements, setting guidelines and defining standardized protocols for single-cell population genetics studies in three separate working groups (1. QC and preprocessing, 2. Cell type classification, 3. eQTL meta-analysis). As soon as this information is ready, guidelines will be shared through our sc-eQTLGen website (https://eqtlgen.org/single-cell.html). Moreover, as a proof of concept, the consortium will summarize all these efforts in a first pilot study in which 3-5 sc-eQTL datasets will be meta-analyzed. This publication is expected to be ready by the end of 2020.

We now specify the expected timeline of our consortium in the “Future clinical implications” section:

“Nevertheless, we expect that most statistical power to pinpoint causal genes will be gained through the other goals of the sc-eQTLGen consortium: the reconstruction of cell type-specific gene regulatory networks (expected by the end of 2022), the mapping of cell type-specific trans-eQTLs and co-expression QTLs (expected by the end of 2021). These efforts will enable us to ascertain how the prioritized cis-eQTL genes (expected by the end of 2021) work together.”

In Box 1:

“By the end of 2020, the sc-eQTLGen consortium will provide standardized pipelines and guidelines for single-cell population genetics studies.”

And in Figure 5 and the corresponding figure legend:

“Figure 5. Deliverables of the single-cell eQTLGen consortium in relation to their future clinical implications. A) In the coming 3 years the sc-eQTLGen consortium aims to deliver 1. Standardized pipelines and guidelines for single-cell population genetics studies (2020); 2. Cell type classification models for PBMCs (2020); 3. Summary statistics of cis- and trans-eQTLs, co-expression QTLs, cell count QTLs and variance QTLs (2021); 4. Reconstruction of personalized, cell type-specific gene regulatory networks (2022). B) These efforts of the consortium will lead to the (1) identification of disease-associated cell types and (2) key disease-driving genes, which together will aid (3) the implementation of personalized medicine and the development of new therapeutics that take all this information into account (cell type- and genotype-specific treatments).”

Further, what type of genotyping will be required for individuals.

The minimum requirements for genotype information are that the genotypes should be assessed through genotype arrays (e.g. Illumina Global Screening Array or ThermoFisher Axiom chip), whole exome or whole genome sequencing. Irrespective of the platform used, we will impute genotypes using the Haplotype Reference Consortium (HRC) imputation reference panel. This will ensure consistency in genotype assignments, preventing potential ambiguity for C/G and A/T SNPs. We realize HRC is confined to SNVs, and plan to look into structural variants at a later stage. Depending on the proportion of cohorts with whole-genome sequencing data in a year from now, we will decide on appropriate strategies for calling structural variants.

We now mention this in Box 1:

“This genetic information can either be efficiently generated using genotype arrays^108^ in combination with imputation-based approaches^109^, or extracted from the scRNA-seq data itself^107, 110^. Within the consortium all reads will be aligned to the *GRCh38/hg38* reference genome and genotypes will be imputed using the Haplotype Reference Consortium reference panel^111^.”

For example, the authors mention that cells from different individuals can be mixed together and "multiplexed" to reduce cost and avoid confounding, but it would be very useful for the authors to show the mapping power increase obtained from multiplexing. Analyses like this may help researchers decide on their preferred collection design which would allow a better harmonization of data generated from outside the consortium.

We apologize for the confusion, but multiplexing samples is not related to the mapping power. Multiplexing allows you to identify doublets from different individuals, thereby allowing a higher number of cells to be loaded (i.e. ‘overloading’) and making the approach more cost-efficient. Moreover, it allows experimental set-ups that combine multiple conditions/individuals together, thereby allowing us to distinguish between experimental and biological effects.

This has previously already been discussed in Box 1:

“First of all, the genetic information that is available for each of the individuals in such cohorts can be used to demultiplex pools of multiple individuals within the same sample.This approach allows to properly randomize experiments, while also significantly reducing cost and confounding effects^23, 107^. This genetic information can either be efficiently generated using genotype arrays^108^ in combination with imputation-based approaches^109^, or extracted from the scRNA-seq data itself^107, 110^. Within the consortium all reads will be aligned to the *GRCh38/hg38* reference genome and genotypes will be imputed using the Haplotype Reference Consortium reference panel^111^. The basic principle behind genetic multiplexing is that enough transcripts harboring SNPs are expressed and detected in each single cell such that cells can be accurately assigned to the donor of origin. Furthermore, as the number of multiplexed individuals increases, the probability that a droplet harbors multiple cells from different individuals increases, thus allowing the detection of multiplets using genetic information. This enables the overloading of cells into standard droplet-based workflows and overall reduction of cost per cell up to about 10-fold (https://satijalab.org/costpercell). As the cost of sequencing and the background multiplet rate reduce, the benefits of multiplexing increase. We anticipate that future workflows will allow for even higher throughput.”

2. The analyses proposed are exciting but the specifics of how they will be run are vague. It would be helpful to catalogue existing tools and identify where new tools are needed, highlighting where the code/algorithms will eventually be found.

The exact details of how all analyses will be run are currently being defined in three separate working groups (1. QC and preprocessing, 2. Cell type classification, 3. eQTL meta-analysis). Each of the working groups are currently benchmarking current approaches for their applicability for large-scale single-cell population studies, and wherever necessary current tools will be adapted for our purpose our new tools will be developed. We expect to have more detailed guidelines ready by the summer of 2020. As soon as this information is ready, guidelines and code will be shared through our sc-eQTLGen website (https://eqtlgen.org/single-cell.html) and through Github. Moreover, as a proof of concept, the consortium will summarize all these efforts in a first pilot study in which 3-5 sc-eQTL datasets will be meta-analyzed. This publication is expected to be ready by the end of 2020.

3. Authors mention gene regulatory networks, when they really plan to study gene co-expression network. "Regulatory" suggests a causal relationship between 2 nodes, while co-expression only relies on correlations. While similar changes in expression levels among cells might suggest a co-regulation, no inference can be made about a regulatory relationship between genes in absence of complementary information such as TF bindings. Some approaches are able to build regulatory networks from expression data, with the addition of prior information (see Sonawane et al., Network Medicine in the age of biomedical big data. 2019. Frontiers in Genetics. doi: 10.3389/fgene.2019.00294).

We apologize for the confusion, but the consortium does want to go beyond reconstruction of gene co-expression networks. In a previously published perspective paper [Van der Wijst et al., 2018 – Genome Medicine], we have explained in detail some of the strategies we would like to take to go from co-expression to gene regulatory networks.

In the “sc-eQTLGen: a federated single-cell eQTL meta-analysis consortium” section, we have now more specifically mentioned that the strategies discussed above, including the use of prior information (e.g. ATAC-seq data, TF binding information), dynamic information (e.g. information extracted from time series data, pseudotime in combination with RNA velocity) and experimental validation (e.g. CRISPR coupled to scRNA-seq), will be applied by the consortium to go from co-expression to gene regulatory networks:

“Finally, the scRNA-seq data will be used to reconstruct GRNs. Two strategies will be explored in the context of sc-eQTLGen. The first approach makes use of the large number of bulk RNA-seq datasets for specific cell types that are available in public RNA-seq repositories^88, 89^. Using this publicly available bulk RNA-seq data, reference co-expression networks will be constructed using cell type-specific data. Subsequently, scRNA-seq data will be used to implement directionality and specify the connections in the network that are affected by specific contexts^44^. The second approach will directly use scRNA-seq data to build cell type-specific GRNs, thereby enabling to immediately take the context-specificity into account. However, the number of genes that can confidently be taken into account by this second approach may be lower due to the sparsity of scRNA-seq data. For both strategies, we will make use of prior information (e.g. ATAC-seq data^58^, TF binding information), dynamic information (e.g. information extracted from time series data^71^, pseudotime^72^ in combination with RNA velocity^45, 46^) and experimental validation (e.g. perturbation experiments^33, 50^) to go from a co-expression to a gene regulatory network.”

4. scRNA-seq data has lower power for eQTL mapping than bulk RNA-seq when matched for sample size. It would be informative for the readers and community to get a better sense of the number of eQTLs that we would expect to map based on individual sample size, number of cells captured by experiment, cell-type proportion in PBMC, etc.

Indeed, with a similar sample size, single-cell data may not necessarily allow you to detect more eQTLs, but it does provide insights that cannot be easily extracted from bulk data (e.g. unbiased detection of cell type-/context-dependent eQTLs, more power to detect co-expression QTLs [Van der Wijst et al., 2018 – Nature Genetics] and being able to extract dynamic information (e.g. pseudotime in combination with RNA velocity) that allows reconstruction of gene regulatory networks [Van der Wijst et al., 2018 – Genome Medicine]). It is these aspects that make single-cell data unique and a valuable approach to pursue to gain additional disease understanding in the future.

Previously, we have performed a small comparison between single-cell and bulk RNA-seq data. For this purpose, we compared the number of *cis*-eQTLs identified in PBMC scRNA-seq data of 45 individuals [Van der Wijst et al., 2018 – Nature Genetics] and whole blood bulk RNA-seq data of 45 individuals (using a random subset of the 2,116 samples that have been reported by Zhernakova et al., Nature Genetics 2017). We subsequently determined what P-Value threshold corresponded to a gene-level FDR of 0.05 (the null distribution was determined using permutations, using the same procedure and software as employed by Zhernakova et al, Nature Genetics 2017). Given this P-Value threshold (P = 8.24x10^-4^) and a sample-size of 45, the corresponding minimal absolute correlation was 0.48 for a *cis*-eQTL to be significant. We then determined what the number of unique genes are with a *cis*-eQTL effect correlation of at least 0.48, as determined in entire bulk RNA-seq cohort of 2,116 samples, to bring down sampling variation. This yielded a set of 2,611 unique genes with a significant *cis*-eQTL effect, indicating that when using the same number of samples, bulk RNA-seq eQTL analysis will yield 6.9x more genes (2,611) with a significant *cis*-eQTL effect, than *cis*-eQTL analysis using scRNA-seq data (379 unique genes with a significant *cis*-eQTL effect).

We now discuss the above in Box 1:

“Even though a single-cell eQTL dataset has less discovery power than an equal-sized bulk RNA-seq eQTL dataset (6.9 fold difference based on the lowest correlation that led to the identification of a significant eQTL from single-cell^16^ vs bulk RNA-seq data^15^), it does provide insights that cannot easily be extracted from bulk data. For example, single-cell data allows for the unbiased detection of cell type- and context-dependent eQTLs and has more power to detect co-expression QTLs^16^. This makes population-based single-cell datasets a valuable addition to bulk-based datasets for studying the effects of genetic variation on gene expression and its regulation^16, 23^.”

However, please note that the analysis above is based on a relatively small number of individuals. Therefore, the exact result will be less accurate. In 2020, we will assess the value of single-cell versus bulk RNA-seq data in much more detail in our first proof of concept paper. In this paper we will perform eQTL, cell count QTL, and variance QTL analyses in 3-5 scRNA-seq datasets (~3-4M cells, ~1,500 individuals) and compare the performance to similar size bulk-RNA seq data.

We now inform the reader about these upcoming analyses while presenting the timeline of our consortium (“Future clinical implications” section and Figure 5):

“Nevertheless, we expect that most statistical power to pinpoint causal genes will be gained through the other goals of the sc-eQTLGen consortium: the reconstruction of cell type-specific gene regulatory networks (expected by the end of 2022), the mapping of cell type-specific trans-eQTLs and co-expression QTLs (expected by the end of 2021). These efforts will enable us to ascertain how the prioritized cis-eQTL genes (expected by the end of 2021) work together.”

5. Many data are mentioned (genomic data, scRNA-seq, scATAC-seq, sc-protein level...), but it is not always clear which ones will be generated, which ones may be generated, and which ones are already existing datasets. Maybe a figure would help?

To be included, every dataset within the consortium will consist of at least scRNA-seq and genotype data. Additional data layers, including CITE-seq and scATAC-seq data, are or will become available for subsets of some of the cohorts. Please also note that the consortium will not necessarily limit itself to the data that is being generated within the consortium. These additional data layers will be useful sources to, for example, further improve our cell type classification or aid in the reconstruction of regulatory networks. However, as large amounts of data are currently still being generated, exact numbers are constantly being updated, and so we cannot be any more specific.

We now further discuss this in the subsection “Gene regulatory network reconstruction”:

“We will extract this prior information from previously published datasets and large-scale efforts, such as ENCODE^2^ and BLUEPRINT^4^. Additionally, we will make use of single-cell information beyond gene expression levels that is or will be collected within subsets of cohorts within the consortium, including information on chromatin accessibility^64, 65^ and expression of enhancers RNAs^66, 67^.”

6. There is limited mention of potential ASE-based or splicing analyses.

ASE-based or splicing analyses are currently not the focus of this consortium. We have several reasons for not doing so at the moment. First, all currently contributing groups have generated either 3’- or 5’-end scRNA-seq data. This type of scRNA-seq data is not very suitable for doing ASE-based or splicing analyses. On the contrary, full-length approaches have been used for this purpose, but are not yet sufficiently cost-efficient so that they can be applied on a very large scale (many cells and many individuals). Nevertheless, technological developments (e.g. Smart-seq3 [M Hagemann-Jensen - ‎2019 bioRxiv]) and ever decreasing sequencing cost may open this avenue to be pursued in the future.

In Figure 5 we now clearly present the deliverables of our consortium, and with that make clear that ASE-based or splicing analyses are not the focus of this consortium (see Figure 5 legend):

“Figure 5. Deliverables of the single-cell eQTLGen consortium in relation to their future clinical implications. A) In the coming 3 years the sc-eQTLGen consortium aims to deliver 1. Standardized pipelines and guidelines for single-cell population genetics studies (2020); 2. Cell type classification models for PBMCs (2020); 3. Summary statistics of cis- and trans-eQTLs, co-expression QTLs, cell count QTLs and variance QTLs (2021); 4. Reconstruction of personalized, cell type-specific gene regulatory networks (2022). B) These efforts of the consortium will lead to the (1) identification of disease-associated cell types and (2) key disease-driving genes, which together will aid (3) the implementation of personalized medicine and the development of new therapeutics that take all this information into account (cell type- and genotype-specific treatments).”

7. There is limited mention of how multi-omics from single cell data may improve GRN or other analyses. There are multiple studies that have now obtained different data modalities from the same cells.

Whilst single-cell multi-omics data will be useful for the consortium in the future, we believe that in its current state it is too time- and cost-consuming to generate sufficient data for personalized GRN reconstruction. However, to acknowledge its potential, we have added the following to the “Gene regulatory network reconstruction” subsection:

‘Additionally, recent advances have made it possible to measure multi-omics data from the very same cell^70, 92^. However, current approaches are very time- and cost-consuming, and therefore limited to only a few hundred cells. As such, currently, this type of single-cell multi-omics data is of limited use for reconstructing personalized GRNs. Nevertheless, as single-cell multi-omics approaches mature, this combined information of gene expression and additional data layers has the potential to improve GRN inference beyond correlating separate omics layers and allows for direct measurements instead.’

8. I would expect some discussion of spatial transcriptomics and its potential.

Initially, the sc-eQTLGen consortium will focus on PBMCs. Specifically for PBMCs spatial transcriptomics data may not be so applicable, as cells are freely floating around in the blood, so spatial organization is more dynamic and cannot be easily captured with current techniques. However, we do acknowledge that for solid tissues and organs this is a highly promising approach that can help us uncover aspects that remain hidden with standard scRNA-seq. Spatial information can, for example, help to disentangle *trans*-eQTL interactions that are modulated through cell-cell communication (e.g. a SNP affects ligand expression in one cell type, and thereby affects downstream receptor signaling in a second cell type [independent of genetics but ligand-downstream receptor signaling: Arneson et al., 2018 - Nature communications; independent of genetics but ligand expression affecting the cell frequency of receptor cells: Smillie et al., 2019 - Cell]). Additionally, this spatial information can be used as prior information to construct more accurate network models that take into account the physical distance between cells and thereby there likelihood of interaction.

We now discuss these aspects in the “Single-cell eQTL analysis: the new era of population genetics” section:

“Another promising avenue that has become available in recent years to gain increased insights in the link between genetics and disease, is through the use of spatial transcriptomics technologies, including MERFISH, seqFISH+, Slide-seq and 10x Visium^34, 35^. While for PBMCs this approach may not be applicable, in solid tissues and organs this extra layer of spatial information is extremely valuable. For example, it can help to disentangle trans-eQTL interactions that are modulated through cell-cell communication (e.g. a SNP affects ligand expression in one cell type, and thereby affects downstream receptor signaling in a second cell type). Despite not having this spatial information available in PBMCs, other approaches that consider receptor-ligand expression pairs do provide insights in potential cell-cell interactions. These approaches have been successfully applied before to uncover how the ligand expression in one cell type can affect the frequency^36^ or the downstream signaling^37^ of another cell type expressing the corresponding receptor.”

And in the “Single-cell GRN reconstruction: taking eQTLs one step further” section:”

“Such priors on regulatory interactions can be derived from, for example, ChIP-seq data^57^, ATAC-seq data^58^, spatial information^34, 35^ or from perturbation experiments^33, 50^.”

9. How does the consortium and its work relate to/differ from the following project?

The sc-eQTLGen consortium does currently not yet have its own funding. Therefore, individual groups generate their own funding through other sources such as the Chan Zuckerberg Initiative (CZI). Jimmie Ye is one of the PIs that is involved in both this CZI project and the sc-eQTLGen consortium. This CZI project covers just parts of the goals of the sc-eQTLGen consortium (identification of context- and cell type-specific eQTLs, but not reconstruction of personalized, context-specific gene regulatory networks) and is limited to a small number of groups. To truly capture all immune phenotypic variation (genetic and environmental), and being able to study *trans*-effects in a genome-wide fashion, a larger collaborative effort (such as sc-eQTLGen) is required. So both projects are related and will benefit from each other, but they will have their own focus points.

10. Please outline the deliverables proposed for the consortium (including a timeline for when they will be available).

The main deliverables for the sc-eQTLGen consortium (focused on single-cell PBMC data initially):

1. Standardized pipelines and guidelines for single-cell population genetics studies (Q2-Q3 2020).

2. Automatic cell type classification model for PBMCs (Q2-Q3 2020).

3. Proof of concept eQTL paper (3-5 cohorts – 3-4M cells - ~1,500 individuals): Summary statistics of cell type- and context-specific *cis*-eQTLs, cell count QTLs, variance QTLs (Q4 2020).

4. Main single-cell eQTL paper (across all cohorts): Summary statistics of genome-wide *cis*- and *trans*-eQTL, co-expression QTL, cell count QTL and variance QTL analyses (Q4 2021)

5. Proof of concept gene regulatory network paper (1 cell type, multiple environmental conditions): Reconstruction of personalized, cell type-specific gene regulatory networks (Q1 2021).

6. Main single-cell gene regulatory network paper (across all cohorts and in all cell types): Reconstruction of personalized, cell type-specific gene regulatory networks, including experimental validation (e.g. using CRISPR perturbation scRNA-seq coupled screens) (Q4 2022).

We now specify the expected timeline of our consortium in the “Future clinical implications” section and in Figure 5:

“Nevertheless, we expect that most statistical power to pinpoint causal genes will be gained through the other goals of the sc-eQTLGen consortium: the reconstruction of cell type-specific gene regulatory networks (expected by the end of 2022), the mapping of cell type-specific trans-eQTLs and co-expression QTLs (expected by the end of 2021). These efforts will enable us to ascertain how the prioritized cis-eQTL genes (expected by the end of 2021) work together.”

And in the figure legend of Figure 5:

“Figure 5. Deliverables of the single-cell eQTLGen consortium in relation to their future clinical implications. A) In the coming 3 years the sc-eQTLGen consortium aims to deliver 1. Standardized pipelines and guidelines for single-cell population genetics studies (2020); 2. Cell type classification models for PBMCs (2020); 3. Summary statistics of cis- and trans-eQTLs, co-expression QTLs, cell count QTLs and variance QTLs (2021); 4. Reconstruction of personalized, cell type-specific gene regulatory networks (2022). B) These efforts of the consortium will lead to the (1) identification of disease-associated cell types and (2) key disease-driving genes, which together will aid (3) the implementation of personalized medicine and the development of new therapeutics that take all this information into account (cell type- and genotype-specific treatments).”

11. Aspects of future data sharing and accessibility are essential to address.

Similarly as we have previously done within the eQTLGen consortium (http://www.eqtlgen.org/), summary statistics (including non-significant findings) will be made available through the sc-eQTLGen website (https://eqtlgen.org/single-cell.html).

This information is now mentioned in the “Future clinical implications” section:

“To increase the impact of this work, all code, guidelines and summary statistics will become freely available to the community through the sc-eQTLGen website (https://eqtlgen.org/single-cell.html). For any additional information, please visit the contact page (https://eqtlgen.org/contact.html).”

12. Please explain how new individuals can become members of the consortium.

Individuals or groups who would like to have additional information or are interested in joining the consortium, can visit the contact page of the sc-eQTLGen website (https://eqtlgen.org/contact.html).

We now mention this in the “Future clinical implications” section:

‘*For any additional information, please visit the contact page (https://eqtlgen.org/contact.html).’*

13. Please explain the consortium will be funded.

Currently, each group is separately funded through their own funding. Based on data generated in the pilot study that will be conducted in 2020, the consortium aims to apply for consortium-wide funding through initiatives like the Chan Zuckerberg Initiative, national (e.g. NIH) or transnational (e.g. H2020) funding.